# Possible Role of Inflammation and Galectin-3 in Brain Injury after Subarachnoid Hemorrhage

**DOI:** 10.3390/brainsci8020030

**Published:** 2018-02-07

**Authors:** Hirofumi Nishikawa, Hidenori Suzuki

**Affiliations:** Department of Neurosurgery, Mie University Graduate School of Medicine, Tsu 514-8507, Japan; trygetter10@yahoo.co.jp

**Keywords:** galectin-3, early brain injury, inflammation, subarachnoid hemorrhage

## Abstract

Aneurysmal subarachnoid hemorrhage (SAH) is known as one of the most devastating diseases in the central nervous system. In the past few decades, research on SAH has focused on cerebral vasospasm to prevent post-SAH delayed cerebral ischemia (DCI) and to improve outcomes. However, increasing evidence has suggested that early brain injury (EBI) is an important mechanism contributing to DCI, cerebral vasospasm as well as poor outcomes. Though the mechanism of EBI is very complex, inflammation is thought to play a pivotal role in EBI. Galectin-3 is a unique chimera type in the galectin family characterized by its β-galactoside-binding lectin, which mediates various pathologies, such as fibrosis, cell adhesion, and inflammation. Recently, two clinical studies revealed galectin-3 to be a possible prognostic biomarker in SAH patients. In addition, our recent report suggested that higher acute-stage plasma galectin-3 levels correlated with subsequent development of delayed cerebral infarction that was not associated with vasospasm in SAH patients. We review the possible role and molecular mechanisms of inflammation as well as galectin-3 in brain injuries, especially focusing on EBI after SAH, and discuss galectin-3 as a potential new therapeutic or research target in post-SAH brain injuries.

## 1. Introduction

Subarachnoid hemorrhage (SAH) by rupture of cerebral aneurysm is an important cause of stroke mortality and morbidity [1]. Recent studies have shown that the inflammatory response, including glial cell activation, contributes to early brain injury (EBI) and delayed cerebral ischemia due to cerebral vasospasm and/or other etiologies, causing poor outcomes after SAH [1]. After SAH, blood components including heme, thrombin, platelets and leukocytes activate microglia, the frontline soldiers of immune defense in the central nervous system (CNS), which serve as tissue-resident macrophages and produce pro-inflammatory cytokines and galectin-3 [1,2]. Recently, matricellular proteins, such as galectin-3, tenascin-C (TNC) and osteopontin, have garnered significant attention as biomarkers for various diseases [2,3,4]. In particular, recommendations for the measurement of galectin-3 are included in the American College of Cardiology Foundation and the American Heart Association guidelines to aid in risk stratification in patients with heart failure [5]. Galectin-3 may be closely linked to the inflammatory cascade, but information as to the role of galectin-3 is limited in the CNS. Thus, this review focuses on possible roles of galectin-3 in brain injuries after SAH, relevant to inflammation and microglia.

## 2. Inflammation Plays a Pivotal Role in EBI after SAH

SAH is known to be one of the most devastating diseases despite advances in diagnosis and treatment. Though there has been increasing attention being focused on EBI in the research of SAH, molecular mechanisms of EBI are very complex, and not well understood [6,7]. EBI is a concept to explain acute pathophysiological events that occur in the brain before the onset of cerebral vasospasm within the first 72 h of SAH [6]. That is, EBI consists of any type of brain insult or pathophysiology other than iatrogenic brain injury, including neuroinflammation, neuronal apoptosis, blood–brain barrier (BBB) disruption, and microcirculatory disturbance [6]. In the acute stage of SAH, the rupture of cerebral aneurysm not only induces transient global cerebral ischemia secondary to elevated intracranial pressure and mechanical stress, but also causes various pathological conditions, such as inflammatory reactions, generation of reactive oxidative stress [8], seizure, and spreading ischemia [9]. Furthermore, various substances produced by SAH, including heme, fibrinogen, intracellular components and inflammation-related proteins, stimulate cell surface receptors including Toll-like receptor (TLR) 4, and activate several inflammatory pathways [10,11]. These pathophysiological processes may cause EBI, which in turn may aggravate neuroinflammation, leading to delayed cerebral ischemia, cerebral vasospasm and/or cerebral infarction [6,10,12]. Thus, early management of inflammation may be an important therapeutic strategy to treat or prevent EBI as well as delayed cerebral ischemia, vasospasm and infarction, and to improve outcomes after SAH [12,13].

## 3. Possible Molecular Mechanisms of Inflammation in EBI

### 3.1. Trigger Factors and Location of Inflammation in EBI

Global cerebral ischemia secondary to elevated intracranial pressure and mechanical stress by an aneurysmal rupture directly cause damages of brain tissues and neuronal cells, leading to apoptosis and inflammation. Simultaneously, various substances, such as heme, fibrinogen, intracellular components and inflammation-related proteins cause and exacerbate inflammatory reactions by the stimulation of cell surface receptors on various brain tissue and vascular cells [10]. Platelet–leukocyte–endothelial cell interactions were also observed in venules at the cerebral surface, 30 min after experimental SAH, inducing early inflammatory and prothrombogenic responses, and contributing to whole brain injury immediately after SAH [14]. Although experimental studies have demonstrated BBB disruption, cell apoptosis and inflammation in the cerebral cortex, hippocampus and brain capillary endothelial cells as causes of neurological impairments [15,16,17], the microglia seem to be an essential element to regulate inflammation [13,18,19,20]. Microglia are resident immune cells in the CNS, serving as tissue-resident macrophages that influence immune responses to tissue injury and repair [21]. There are two phenotypes of microglia in inflammation-associated neurological injuries: pro-inflammatory phenotype (M1) and anti-inflammatory phenotype (M2) [22,23]. The M1 phenotype tends to release pro-inflammatory cytokines that aggravate tissue injury, such as tumor necrosis factor (TNF)-α, interleukin (IL)-1β, IL-12 and nitric oxide (NO), while the M2 phenotype is likely to release anti-inflammatory cytokines and neurotrophic factors that promote inflammatory resolution and tissue repair [24,25]. However, few studies have been conducted as to phenotypes of microglia after SAH. 

### 3.2. Involved Receptors in EBI

As to cell surface receptors, TLR4 has been identified as one of major receptors promoting inflammation in EBI [10]. TLR4 is expressed on various cell types in CNS, including microglia, neurons, astrocytes, capillary endothelial cells, endothelial and smooth muscle cells of cerebral arteries, as well as peripheral blood cells, such as leukocytes, macrophages and platelets [26]. TLR4 requires its extracellular binding partners, myeloid differentiation factor-2 (MD-2), and cluster of differentiation 14 (CD14), to mediate signal transduction induced by ligands [10]. Then, TLR4 interacts with two distinct adaptor proteins: myeloid differentiation primary-response protein (MyD88) and toll receptor associated activator of interferon (TRIF) [10]. Hanafy reported that when TLR4 knockout mice were exposed to experimental SAH through a cisternal blood injection, neuronal apoptosis was largely TLR4-MyD88-dependent and microglia-dependent in the early phase of SAH, and TLR4-TRIF-dependent and microglia-independent in the late phase of SAH [18]. The MyD88-dependent pathway activates a transcriptional factor nuclear factor (NF)-κB and produces pro-inflammatory cytokines or mediators, such as TNF-α, ILs (IL-1β, IL-6, IL-8, and IL-12), intracellular adhesion molecule-1, monocyte chemoattractant protein-1, matrix metalloproteinase (MMP)-9, cyclooxygenases, and reactive oxygen species (NO, hydrogen peroxide, and superoxide) [10,26]. The MyD88-dependent pathway also activates another transcription factor, activator protein-1, which is mainly mediated by mitogen-activated protein kinases (MAPKs), such as c-Jun N-terminal kinase (JNK), p38 and extracellular signal-regulated kinase (ERK) [27]. Yan et al. reported that a selective inhibition of MyD88 alleviated SAH-induced inflammatory responses and apoptosis through the downregulation of MAPKs and NF-κB signaling pathways [28].

As to other receptors, it was reported that vascular endothelial growth factor (VEGF) exacerbated post-SAH BBB disruption through VEGF receptor (VEGFR)-2 in an endovascular perforation SAH model of mice, and that the direct blockage of VEGF pathways by anti-VEGF or anti-VEGFR-2 antibodies alleviated post-SAH BBB disruption [29]. In addition, a selective blockage of platelet-derived growth factor receptor by imatinib mesylate prevented neuronal apoptosis as well as activation of MAPKs and upregulation of TNC, a matricellular protein, in an experimental SAH rat [17]. These receptors aggravated EBI such as BBB disruption and neuronal apoptosis through the activation of MAPK pathways and the interaction with TNC [17,29]. Overexpression of peroxisome proliferator-activated receptor β/δ, which is a ligand-activated transcriptional factor belonging to the nuclear hormone receptor family, was reported to ameliorate EBI by improvement of brain edema, BBB disruption and neuronal apoptosis associated with the downregulation of NF-κB and MMP-9 [30]. Future studies would reveal the involvement of other receptors and signaling pathways in EBI [31,32,33]. 

### 3.3. Major Inflammatory Signaling Pathways in EBI

MAPK pathway was found to play a crucial role in EBI, and MMP-9 was identified as one of the important downstream effectors related to BBB permeability and neuronal apoptosis [12,16]. MAPK signaling pathways regulate gene expression in eukaryotic cells, being linked to extracellular signals [34]. There are four well characterized and widely-studied MAPK pathways: ERK1/2 MAPK; p38 MAPK; JNK MAPK; and ERK5 MAPK [34]. These pathways are activated through phosphorylation by upstream kinases, recruited via diverse extracellular signal events [34]. In SAH, the ERK1/2 MAPK pathway was reported to regulate cerebral blood flow through a variety of mechanisms, such as inflammatory responses [34,35], while the p38 MAPK pathway in the cortex and arterial walls was activated to induce post-SAH neuronal cell death, endothelial cell apoptosis, and inflammatory cytokine expression, leading to EBI and delayed cerebral vasospasm [34,36,37]. In addition, the JNK MAPK pathway was also activated in the process of post-SAH brain injury [34,38]. These MAPK pathways do not have entirely independent functions, and sometimes there is cross talk among these pathways [16,17,34]. 

The NF-κB signaling pathway was also reported to play a pivotal role in EBI [30,32,39]. NF-κB is a well-known transcriptional factor which induces MMP-9 expression with the effect of proapoptosis [40], and regulates the expression of various genes encoding pro-inflammatory mediators [41,42]. In normal cells, NF-κB is present in the cytoplasm as an inactive heterodimer composed of two subunits: p50 and p65. The heterodimer forms a complex with an inhibitory protein: IκB-α. When cells are activated by certain inflammatory agents, the IκB-α protein gets degraded rapidly by its phosphorylation, and NF-κB becomes dissociated from IκB-α, being translocated to the nucleus, where it binds to the specific deoxyribonucleic acid sequence present in the promoters of numerous target genes [39]. NF-κB regulates the expression of various components of the immune system, including pro-inflammatory cytokines, chemokines, adhesion molecules, and inducible enzymes, such as cyclooxygenase-2 and inducible NO synthase, as well as proteins that regulate the specific immune response, and control lymphocyte proliferation and differentiation, such as IL-2, IL-12, and interferon (IFN)-γ [39]. 

BBB is mainly composed of microvascular endothelial cells with tight junctions, and astrocytes play a fundamental role in brain hemostasis, regulating the entry of intravascular molecules into the brain. Degradation of zonula occludens-1 and occludin, which are members of tight junction proteins, was reported to cause tight junction opening and BBB permeability in the early stages of SAH [16,43]. BBB disruption causes brain edema as well as greater influx of blood-borne cells and substances into brain parenchyma, thus exacerbating neuroinflammation and brain injuries. However, inhibition of inflammatory signaling pathways would alleviate both BBB permeability and neuronal apoptosis after SAH.

## 4. Galectin-3 Belongs to the Galectin Family, Exhibiting Unique Characteristics

Galectins are composed of more than 15 members of the β-galactoside-binding lectin superfamily, defined by their conserved peptide sequence elements in the carbohydrate-recognition domain (CRD), which are crucial for their affinities to β-galactoside-containing carbohydrate moieties of glycoconjugates [44]. Based on their structures, galectins are classified as proto-type (galectins-1, 2, 5, 7, 10, 11, 13–20), which includes monomers or homodimers with one CRD; chimera type (galectin-3), which contains a non-lectin part made of proline- and glysine-rich short tandem repeats connected to a CRD; and tandem-repeat type (galectins-4, 6, 8, 9, 12), which contains two CRDs with different binding partners, connected by a single polypeptide chain (Figure 1) [44,45]. Galectins have been described as regulators of a wide variety of biological processes, including immune response, metabolism, signal transduction, cell-cell/extracellular matrix interaction, and apoptosis [45]. Galectins are distributed both inside and outside cells and are believed to have roles in both intra- and extracellular milieus. Extracellular galectins exhibit bivalent or multivalent interactions with glycans on cell surfaces, leading to various cellular responses, including production of cytokines and other inflammatory mediators, cell adhesion, migration, and apoptosis [44]. On the other hand, intracellular galectins can function in signaling pathways and alter biological responses, including apoptosis, cell differentiation, and cell migration [44].

Galectin-3, the only chimera-type galectin, has been found in the nucleus and cytoplasm, at the cell surface, and in the extracellular fluid surrounding several cell types [46]. Galectin-3 comprises a CRD and N-terminal non-CRD for carbohydrate binding and increases self-association, respectively. This allows galectin-3 to bridge effectively between different ligands and to exert different functions [46,47,48]. In addition, galectin-3 has been reported to bind to a number of intracellular proteins, which are known to participate in intracellular signaling pathways [44]. In recent decades, the literature on galectin-3 has been rapidly growing, and has revealed that galectin-3 plays diverse roles in the inflammation of the CNS, combining its pro-inflammatory role with its remodeling capacity in damaged neuronal tissues [49].

## 5. Molecular Mechanisms and Clinical Implication of Galectin-3 in Systemic Diseases

Galectin-3 is a ligand, activated by binding to oligosaccharides, including various glycosylated matrix proteins, such as laminin, collagen, elastin, fibronectin, integrin and TNC, via its CRD [50]. While the cellular actions of galectin-3 are to mediate cell adhesion, proliferation and fibrosis, increasing evidence has demonstrated that galectin-3 can also activate various types of immune-associated cells, including neutrophils, monocytes, dendritic cells, macrophages, and mast cells [44,51]. In line with these, galectin-3 has been implicated in various systemic diseases, such as cancer [52,53], renal and cardiac fibrosis [54,55,56], and immunological disorders [57,58,59,60].

The interaction between galectin-3 and cell adhesion molecules, such as integrin, cadherin and mucin 1, regulates the invasion and metastasis of cancer cells [61]. Markowska et al. reported that galectin-3 modulated VEGF- and basic fibroblast growth factor-mediated angiogenesis [62]. The CRD of galectin-3 binds to N-acetylglucosaminyltransferase V-modified N-glycans on αvβ3 integrin and activates focal adhesion kinase-mediated signaling pathways that modulate endothelial cell migration in the angiogenic cascade [62]. In this pathology, the CRD of galectin-3 is important for cell surface binding and internalization of galectin-3 by the endothelial cells [53].

A correlation between galectin-3 expression and fibrosis has been found in several pathologies, such as liver, renal, idiopathic pulmonary, and cardiac fibrosis [54,55,63,64,65]. Macrophages can secrete galectin-3 in the extracellular space and activate resting fibroblasts into a matrix-producing phenotype [64,65]. As to cardiac fibrosis, aldosterone has been demonstrated to induce galectin-3 secretion by macrophages, via the phosphoinositide 3-kinase/Akt and NF-κB signaling pathways, through mineralocorticoid receptors [66]. Although the precise mechanisms through which galectin-3 mediates extracellular matrix remodeling and fibrosis remain unclear, activation of the Janus kinase (JAK)/signal transducer and activation of transcription (STAT) and protein kinase C pathways, as well as oxidative stress and inflammation, have been suggested [63]. On the other hand, modified citrus pectin (MCP), which is known to be a galectin-3 inhibitor, attenuated cardiac dysfunction and fibrosis by reducing myocardial inflammation and fibrogenesis in rats and mice [55,65]. In clinical settings, plasma galectin-3 levels are increased and considered to be a strong prognostic biomarker in patients with heart failure [67]. The Prevention of REnal and Vascular ENd-stage Disease (PREVEND) cohort study, including 7968 individuals, demonstrated that higher plasma galectin-3 was associated with increasing age and risk factors of cardiovascular diseases, especially in females, and independently predicted mortality in the general population [54].

## 6. Galectin-3 Activates Microglia and Inflammatory Reactions in CNS

In the development of CNS, galectin-3 is expressed in a variety of glial cells, including microglia, astrocytes and oligodendrocytes [68], and contributes to the migration of neuroblasts, oligodendrocyte myelination and neurite outgrowth [49,69]. Meanwhile, increasing studies have suggested that galectin-3 released by activated microglia in response to pro-inflammatory stimuli seems to participate in brain immune responses (Figure 2) [2,49,70]. Jeon et al. demonstrated that the expression or secretion of galectin-3 into the extracellular matrix was enhanced in glia under IFN-γ stimulated, inflamed conditions, and that exposure of galectin-3 to glia resulted in the production and upregulation of pro-inflammatory cytokines, such as IL-1β, and IL-6, through the activation of the JAK-STAT and NF-κB pathways [46]. In a study using murine global brain ischemia or neuroinflammatory models with an intranigral injection of lipopolysaccharide, galectin-3 released by microglia acted as an endogenous paracrine TLR4 ligand through its CRD, and depletion of galectin-3 exerted neuroprotective and anti-inflammatory effects [48]. In addition, in a controlled cortical impact mouse model of head injury, the upregulation of galectin-3 in microglia and the released form of galectin-3 in the cerebrospinal fluid were observed at 24 h after head injury, and binding between galectin-3 and TLR4 was confirmed [70]. Administration of neutralizing antibodies against galectin-3 attenuated the expression of IL-1β, IL-6, TNF-α and NO synthase 2 and exerted neuroprotection in the cortical and hippocampal cell populations after head injury [70]. On the other hand, there are some reports suggesting galectin-3’s protective effects. Lalancette et al. demonstrated that targeted deletion of galectin-3 exacerbated ischemic damage and neuronal apoptosis after cerebral ischemia through the reduction of interaction between galectin-3 and insulin-like growth factor (IGF)-1, leading to downregulation of IGF-1 and suppression of resident microglia activation and proliferation in response to ischemic injury [47]. Wesley et al. conducted an in vitro study, in which galectin-3 contributed to angiogenesis and microglia migration via integrin-linked kinase signaling pathways, suggesting galectin-3’s implication in post-ischemic repair [71]. Other in-vitro and in-vivo studies showed that galectin-3 was expressed in a delayed fashion by activated microglia/infiltrating macrophages and astrocytes in a rat ischemic brain, and might play a role in post-ischemic tissue remodeling by enhancing angiogenesis and neurogenesis [72]. Pasquini et al. reported that glia-derived galectin-3 promoted oligodendroglia differentiation, contributing to myelin integrity and function with critical implications in the recovery of inflammatory demyelinating disorders [68]. These seemingly conflicting findings may be due to diverse functions and expression sites of galectin-3 that may have different effects depending on the nature of the disease condition and the time since injury [2].

## 7. Clinical Implication of Galectin-3 in CNS

Clinically, concentrations of plasma galectin-3 may be useful as a biomarker in some CNS diseases. Yan et al. reported that elevated plasma galectin-3 levels were strongly associated with inflammation, severity and poor outcomes in patients with acute intracerebral hemorrhage [73]. In 100 patients with severe traumatic brain injury, plasma galectin-3 concentrations had a close relationship with inflammation, trauma severity and clinical outcomes [74]. Meanwhile, a study involving 8444 participants from the general population-based FINRISK 1997 cohort demonstrated that galectin-3 was weakly associated with incident ischemic stroke over a 15-year follow up period, but this relationship was attenuated in multivariate adjusted models, though galectin-3 levels were predictive for future cardiovascular events, such as all-cause mortality, cardiac death and heart failure [75]. 

As for SAH, Liu et al. demonstrated that higher plasma galectin-3 levels had a significant correlation with worse admission neurological status and more subarachnoid and intraventricular hematoma volume on computed tomography scans [76]. In addition, galectin-3 independently predicted 6-month clinical outcomes, suggesting their potential to be a good prognostic biomarker [76]. Our recent study showed that higher plasma galectin-3 levels on days 1–3 post-SAH were correlated with 3-month poor outcomes in non-severe aneurysmal SAH patients [77]. Interestingly, higher acute-stage plasma galectin-3 levels were associated with subsequent development of delayed cerebral ischemia and cerebral infarction, but not cerebral vasospasm [77].

## 8. Galectin-3 Is a Novel Target for the Research of EBI

Although high plasma galectin-3 levels seem to correlate with the severity of SAH, subsequent development of delayed cerebral infarction without cerebral vasospasm, and poor outcomes [76,77], the functional role and molecular mechanisms of galectin-3 in SAH remain unclear. As described above, experimental studies of other diseases showed that galectin-3 was secreted by glial cells, acted as a TLR4 ligand, and activated microglia, inducing the upregulation of pro-inflammatory cytokines [47,48,70]. According to recent studies, inflammation also plays a pivotal role in EBI [78], and TLR4 may be a key receptor for initiating and promoting EBI [10]. Our preliminary study using endovascular perforation SAH models in mice revealed that the inhibition of galectin-3 by MCP in an acute stage attenuated post-SAH brain edema and BBB disruption, leading to improvement of neurological scores. The mechanisms of MCP for preventing BBB disruption seemed to include the inhibition of inflammatory reactions through the downregulation of JAK-STAT pathways, but further high-quality studies are needed to confirm the findings [79]. In addition, there are little data available as to the functional significance of galectin-3 in a clinical setting, especially in SAH, which is also an important issue to be solved. However, available data suggest that elucidating the role of galectin-3 in EBI would provide new insight into the mechanisms of EBI, and a novel therapeutic approach for EBI or neuroinflammation to prevent delayed cerebral infarction and to improve outcomes after SAH. Thus, further studies regarding galectin-3 are expected.

## Figures and Tables

**Figure 1 brainsci-08-00030-f001:**
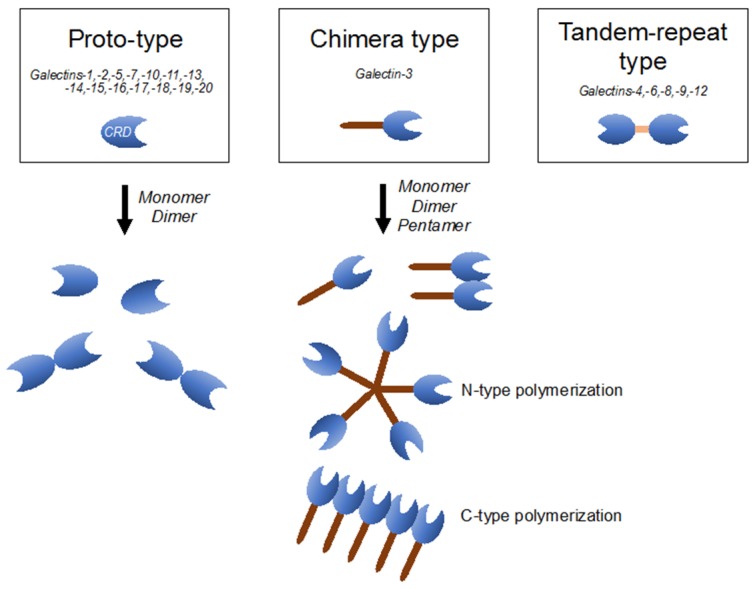
Classification of the galectin family is defined according to the respective carbohydrate-recognition domain (CRD) structure. Proto-type one consists of monomers or homodimers with one CRD; chimera-type one has a long non-lectin domain and one CRD; and tandem-repeat type one contains two distinct CRDs.

**Figure 2 brainsci-08-00030-f002:**
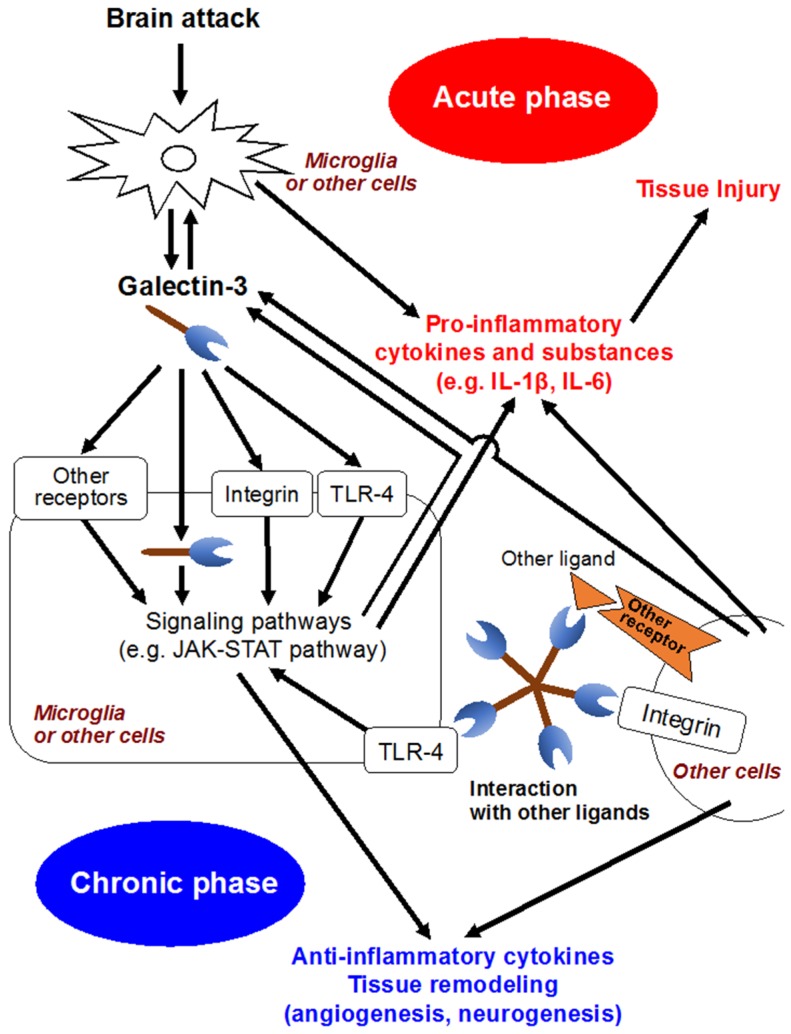
Possible molecular mechanisms of galectin-3. IL, interleukin; JAK-STAT, Janus kinase/signal transducer and activation of transcription; TLR4, Toll-like receptor 4.

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
