# Peer review of "Possible Role of Inflammation and Galectin-3 in Brain Injury after Subarachnoid Hemorrhage"

_brainsci, 2018, doi:10.3390/brainsci8020030_

Reviewer 1 Report

The authors provide a wide ranging review of the action of galectin-3 in subarachnoid haemorrhage. They present some interesting and plausible findings postulating a specific role in SAH, however the majority of the article is based on pre-clinical models. Other than correlative study (that do necessarily imply a mechanistic role for galectin-3) in TBI and SAH patients, there is very little available in the human literature to draw firm conclusions. I think this needs to be made clear in an otherwise good review. As such the conclusions are still somewhat speculative.

Author Response

We would like to thank you for your encouraging comments. According to your suggestions, we modified the last paragraph on lines 266-283, page 7. The modified sentences or phrases in our manuscript are shown in red.

Reviewer 2 Report

The present manuscript is a review on the role of inflammation and galectin-3 in subarachnoid hemorrhage. Overall, the review is well written describing current knowledge on the topic. Here are few suggestions to improve the manuscript:

1.  It will be helpful to have and introductory paragraph to briefly describe significance of inflammation, galectin-3 and microglia in SAH and why the review focuses on these factors.

2. Both figures need major revision. In the current form, I found both pictures unhelpful and confusing.

Minor suggestions:

- line 69: change “ … myeloid differentiation factor 2 and ….” to “…myeloid differentiation factor 2 (MD-2) and…”

- line 69: change “…cluster of differentiation 14 to mediate…” to “…cluster of differentiation 14 (CD14) to mediate…”

-line 194: should be “ females”

Author Response

We would like to thank you for your encouraging comments. We have improved our manuscript according to your suggestions, and the modified sentences or phrases in our manuscript are shown in red.

1. According to your suggestions, we added the paragraph subtitled "Introduction" on lines 24-38, page 1.

2. According to your suggestions, we modified the both figures, and the figure legends appropriately.

Minor suggestions:

        According to your suggestions, we modified the words or phases appropriately.